# Evolving Risk of Acute Kidney Injury in COVID-19 Hospitalized Patients: A Single Center Retrospective Study

**DOI:** 10.3390/medicina58030443

**Published:** 2022-03-18

**Authors:** Fahad D. Algahtani, Mohamed T. Elabbasy, Fares Alshammari, Amira Atta, Ayman M. El-Fateh, Mohamed E. Ghoniem

**Affiliations:** 1Department of Public Health, College of Public Health and Health Informatics, University of Ha’il, Ha’il 2440, Saudi Arabia; tharwat330@gmail.com; 2Department of Health Informatics, College of Public Health and Health Informatics, University of Ha’il, Ha’il 2440, Saudi Arabia; f.alhammzani@uoh.edu.sa (F.A.); am.atta@uoh.edu.sa (A.A.); 3Department of Internal Medicine, AL-AHRAR Teaching Hospital, Zagazig 44919, Egypt; ayman191284@yahoo.com; 4Department of Internal Medicine, College of Medicine, University of Ha’il, Ha’il 2240, Saudi Arabia; mo.ghonim@uoh.edu.sa or; 5Department of Internal Medicine, Faculty of Medicine, Zagazig University, Zagazig 44519, Egypt

**Keywords:** coronavirus disease (COVID-19), acute kidney injury (AKI), sequential organ failure assessment (SOFA)

## Abstract

*Background and Objectives*: Within a year, COVID-19 has advanced from an outbreak to a pandemic, spreading rapidly and globally with devastating impact. The pathophysiological link between COVID-19 and acute kidney injury (AKI) is currently being debated among scientists. While some studies have concluded that the mechanisms of AKI in COVID-19 patients are complex and not fully understood, others have claimed that AKI is a rare complication of COVID-19-related disorders. Considering this information gap and its possible influence on COVID-19-associated AKI management, our study aimed to explore the prevalence of AKI and to identify possible risk factors associated with AKI development among COVID-19 hospitalized patients. *Materials and Methods*: A retrospective cohort study included 83 laboratory-confirmed COVID-19 patients hospitalized at the isolation department in a tertiary hospital in Zagazig City, Egypt between June and August 2020. Patients younger than 18 years of age, those diagnosed with end-stage kidney disease, or those on nephrotoxic medications were excluded. All study participants had a complete blood count, liver and renal function tests, hemostasis parameters examined, inflammatory markers, serum electrolytes, routine urinalysis, arterial blood gas, and non-enhanced chest and abdominal computer tomography (CT) scans. *Results*: Of the 83 patients, AKI developed in 24 (28.9%) of them, of which 70.8% were in stage 1, 8.3% in stage 2, and 20.8% in stage 3. Patients with AKI were older than patients without AKI, with hypertension and diabetes being the most common comorbidities. Risk factors for AKI include increased age, hypertension, diabetes mellitus, and a higher sequential organ failure assessment (SOFA) score. *Conclusions*: AKI occurs in a considerable percentage of patients with COVID-19, especially in elderly males, those with hypertension, diabetes, and a higher sequential organ failure assessment (SOFA) score. Hence, the presence of AKI should be taken into account as an important index within the risk spectrum of disease severity for COVID-19 patients.

## 1. Introduction

COVID-19 essentially shows as acute respiratory distress with interstitial and alveolar pneumonia; however, it can influence numerous organs, for example, the kidney, heart, digestive tract, blood, and nervous system [1]. The virus may pass through the blood after it infects the lung, gather in the kidney, and cause harm to renal cells. Some affected renal cells may advance to renal failure and may require dialysis as a way of management [2].

According to a recent meta-analysis, the pooled incidence of AKI in COVID-19 was only 8.4%, and its occurrence was linked to a higher mortality rate [3]. AKI, on the other hand, may be as common as 20% in a recent cohort from the New York City area [4]. However, others have reported that AKI was uncommon in COVID-19 and SARS-CoV-2 infection did not result in AKI, or aggravate chronic kidney disease (CKD) in the COVID-19 patients [5]. Despite this, only a few studies have identified the risk factors for AKI in COVID-19 to date. These data are critical for understanding disease onset and progression. Identifying the potential risk factors for AKI in individuals with coexisting COVID-19 should help to guide therapeutic methods and, as a result, improve the prognosis of these patients [6].

Hence, it is important to explore and determine the demographics, clinical, laboratory, and radiologic features of COVID-19-associated AKI among patients as a way of understanding the underlying process between COVID-19 and AKI.

## 2. Materials and Methods

An observational single-center study including (83) laboratory-confirmed COVID-19 patients (age ≥ 18 years), hospitalized at the isolation department in a tertiary hospital in Zagazig City, Egypt from June to August 2020. All studied patients were suspected as COVID-19 patients as they had fever, dry cough, tiredness, diarrhea, loss of taste and/or smell. Accordingly, they were isolated in the medical departments. All of them had mild to moderate symptoms of COVID-19, but no evidence of severe pneumonia such as a SpO_2_ of less than 90% on room air. None of them were admitted to the ICU or underwent mechanical ventilation. According to the WHO guidelines, a confirmed diagnosis of COVID-19 was defined as a positive result of a quantitative real-time reverse-transcriptase-polymerase chain reaction (RT-PCR) detection in a nasopharyngeal swab [7]. Patients younger than 18 years, those previously diagnosed with chronic kidney disease before admission, and those taking any nephrotoxic medications a week prior to admission were excluded from the study.

In order to maintain and ascertain the rights of patients without jeopardizing their privacy and confidentiality, informed verbal consents, in lieu of written informed consents, were obtained directly from all patients or their relatives. The requirements to obtain written informed consents were not attainable due to the inherent hazardous and contagious nature of the COVID-19 epidemic, the strict restrictions by the hospital management, and the urgency of collecting the data. The research protocol was authorized by the University of Ha’il’s institutional review boards under project number (COVID-1927), and was registered at www.clinicalTrials.gov (accessed on 23 July 2021) (NCT04874779).

SARS-CoV-2 RT-PCR testing was performed on nasopharyngeal swabs using Thermofisher SCIENTIFIC’s TaqPathTM COVID 19 CE IVD RT PCR Kit (ThermoFisher Scientific, Waltham, MA, USA), 1000 reactions (Cat. No. A48067). The QUIAGEN Extraction Kit was used to extract viral RNA. Using Quick Dx Applied Biosystems 7500 real-time PCR instruments, the purified nucleic acid was reverse transcribed into cDNA and amplified using the TaqPathTM COVID 19 RT PCR Kit (ThermoFisher Scientific, Waltham, MA, USA) in one stage. Probes annealed to three unique SARS-CoV-2 target sequences: ORF1ab, nucleocapsid (N), and spike (S) primers/probes for bacteriophage MS2. The result was deemed null because two of the three genes and the MS2 (positive control) were all positive. A COVID-19 infection was diagnosed whenever a positive RT-PCR result was identified.

The patients’ medical history, underlying diseases, medications administered at home and in hospital, laboratory data and non-enhanced chest and abdominal computer tomography (CT) scans were retrieved from electronic medical records. According to the Ministry of Health and Population (MOHP), Egypt Management Protocol for COVID-19 Patients [8], version 1.4, 30 May 2020, all patients received standard of care treatment including (a) oxygen supply to achieve SaO_2_ ≥ 90%; (b) hydroxychloroquine (400 mg twice in first day then 200 mg twice for 6 days); (c) vitamin C (1000 mg daily); (d) zinc (50 mg daily); (e) acetylcysteine 200 mg t.d.s.; (f) lactoferrin one sachet (100 mg) twice daily; and (g) anticoagulation (Rivaroxaban 20 mg daily) if D-dimer > 1000.

Laboratory data included complete blood count by automated blood counter, liver and renal function tests by colorimetric method, examination of hemostasis parameters, inflammatory markers, serum Na, serum K, serum Ca, urine dipstick test, and arterial blood gas analysis.

AKI was identified according to the guidelines of the Kidney Disease Improving Global Outcomes (KDIGO). It is defined as any of the following: (I) serum creatinine increase ≥ 0.3 mg/dL (≥26.5 μmol/L) within 48 h; or (II) serum creatinine increase to ≥1.5 times baseline within the previous seven days; or (III) a urine volume ≤ 0.5 mL/kg/h for 6 h. AKI was staged for severity according to the criteria presented in (Table 1) [9]. Accordingly, patients were classified into two groups: AKI and non-AKI groups. Due to a lack of daily urine measurement in the electronic chart, urine output criteria were not used consistently for the diagnosis of AKI. Additionally, we were unable to obtain urinalysis results prior to admission and, subsequently, we could not confirm whether or not proteinuria and hematuria were present before admission.

The Sequential Organ Failure Assessment (SOFA) score was generated in a consensus meeting of the European Society of Intensive Care Medicine in 1994 and was later amended in 1996 [10]. It was developed as a morbidity severity score to focus on organ dysfunction, which can reliably predict disease severity and outcome. It includes six variables, each representing an organ system. Each organ is graded from 0 (normal) to 4 (the most abnormal), providing a score (Table 2).

### Statistical Analysis

Statistical analysis was carried out using the Statistical Package for the Social Science (SPSS, Chicago, IL, USA) version 26 for Microsoft Windows, USA. Demographic and laboratory data were expressed as means ± standard deviations (SD). Categorical variables were presented as frequency and percentage. A Pearson correlation test was performed to determine the relationship between the parameters. For nonparametric data (clinical characteristics of participants), a chi-square test (significance level: *p* < 0.05) was used. One-way analysis of variance (ANOVA) was performed to determine the significant differences in parametric data between the control and treatment groups (Groups I and II). Given the small but equal sample sizes in the control and treatment groups, ANOVA was the preferred test of choice as it remains robust to deviations from normal [11]. We studied factors associated with AKI by univariate logistic regression and reported Odds ratios (OR) and their 95% confident intervals (CI).

## 3. Results

The baseline characteristics of patients are shown in (Table 3). A total of 83 patients, 46 (55.4%) of whom were male with mean age (years) of (52.9 ± 12.4) and the mean of their mean arterial pressure (mmHg) was (90.1 ± 5.4). While only 21 patients (25.3%) had no underlying disease, the rest were diagnosed with comorbidities such as hypertension (*n* = 23, 27.7%), diabetes mellitus (*n* = 19, 22.9%), cardiovascular diseases (*n* = 10, 12.1%), chronic obstructive pulmonary disease (*n* = 7, 8.4%), and liver disease (*n* = 3, 3.6%). Chest computed tomography scan (non-enhanced) examination on hospital admission was conducted and demonstrated bilateral peripheral ill-defined patchy ground-glass opacities in both lower lung fields (Figure 1).

All patients were evaluated using the KDIGO classification: AKI occurred in 24 patients (28.9%). In our study, AKI group patients were significantly older than the non-AKI group (57.71 ± 11.0 vs. 51.07 ± 12.5, *p* = 0.026), respectively, and AKI was found to be more prevalent among male patients (58.3%). Among the AKI group, 17 patients (70.8%) were in stage 1, two patients (8.3%) were in stage 2, and five patients (20.9%) were in stage 3 (Figure 2). Only one 62-year-old-male patient with AKI was in stage 3, with serum creatinine of 12.4 mg/dL. After he had undergone two sessions of hemodialysis, his serum creatinine returned to 5.6 mg/dL. Hypertension and diabetes were the most coexisting diseases and were significantly higher in the AKI group than in the non-AKI group (33.3% vs. 25.4%) and (29.2% vs. 20.3%), respectively. A computerized tomography (non-enhanced) scan of the abdomen in AKI group demonstrated perinephric fat stranding with irregularity of the outer border of the kidney (Figure 3). There was no statistically significant difference in symptoms between the AKI and non-AKI patients.

The baseline laboratory data of patients is shown in (Table 4). When compared to the non-AKI group, AKI group had a higher serum Creatinine (2.5 ± 1.9 vs. 1.5 ± 0.8, *p* ≤ 0.007); lower PaO_2_/FiO_2_ ratios (407.28 ± 28.83 vs. 414.07 ± 31.75, *p* ≤ 0.046); and higher sequential organ failure assessment (SOFA) (1.88 ± 0.95 vs. 1.33 ± 0.66, *p* ≤ 0.004). Routine urinalysis with a dipstick test was performed on all the study participants and proteinuria was found among 26 (31.3%) patients and hematuria among 16 (19.3%) patients, thus hematuria and proteinuria were more prevalent among the AKI (45.8%, 25%) group when compared with the non-AKI group (25.4%, 16.9%), respectively. Other laboratory results showed no statistically significant difference between the two groups.

We studied various baseline characteristics as potential risk factors of AKI in COVID-19 as detailed in (Table 5). Univariate logistic regression demonstrated that increased age, ((OR): 2.051, (95% CI): 1.771–2.373, *p* ≤ 0.001), hypertension ((OR): 1.612, (95% CI): 1.424–1.823, *p* ≤ 0.002), diabetes mellitus ((OR): 1.094, (95% CI): 1.032–1.24, *p* ≤ 0.012), and higher SOFA score ((OR): 1.051, (95% CI): 1.012–1.218, *p* ≤ 0.023) were independent risk factors for the evolution of AKI among COVID-19 patients (Figure 4). During the study era, there was no mortality among COVID-19 patients.

## 4. Discussion

Our study found an incidence of AKI (28.9%) among patients admitted with COVID-19 at the isolation department in a tertiary hospital, Zagazig, Egypt. The incidence of AKI among patients with COVID-19 varies across existing studies [12,13,14], which can be attributed to race, patient characteristics, variation in fluid hemodynamic management, medication use, and timing of hospital admission. Contrary to our results, Wang et al. (2020) [5] and Guan et al. (2020) [15] failed to report any obvious azotemia in COVID-19 patients that could be due to relatively younger age and a larger number of study participants.

Although the exact pathophysiology of AKI in COVID-19 is still obscure, we can presume that fever or decreased fluid intake may play a significant role. Some researchers have supposed that dehydration may lead to a reduction in glomerular filtration rate and pre-renal AKI [15].

Although, we did not observe any statistically significant difference in inflammatory markers between the studied groups, others have supported the evolution of cytokine storm syndrome (CSS), possibly due to virus-induced sepsis associated with cytokine release into the circulation, systematic inflammatory responses, and cardiomyopathy with a propensity to cardiorenal syndrome type 1 headway [16]. In addition, organ crosstalk such as CSS cardiomyopathy and acute viral myocarditis could be involved in hypotension, renal hypoperfusion, and a reduction in glomerular filtration rate [17]. Indeed, rapid viral replication may cause massive endothelial cell death, triggering the production of different cytokines such as TNF-α [18], IFN-γ-induced cytokine cycling [19], or overproduction of IL-6 with lung–kidney bidirectional damage [20]. Some studies have detected complement C5b-9 (membrane attack complex “MAC complex”) deposition on tubules and infiltration of CD68+ macrophages in the tubule-interstitium by immunofluorescence examination of renal tissues [21]. Conversely, previous reports [22,23] did not detect any correlation between complement activation and AKI.

Our study reports an increased D-dimer levels in the AKI group that somewhat points to circulating microthrombi formation, which is congruent with previous studies indicating that COVID-19 patients are at risk for disseminated intravascular coagulation and thrombosis of both arterial and venous systems [24] with subsequent right heart failure and development of AKI [25]. These are in line with Post et al. (2020) [26], who proposed kidney infarction as a suspected cause of AKI through direct viral induced antiphospholipid antibody production, leading to thromboembolic events [27]. Our findings were further supported by a recent post-mortem histopathologic analysis of 26 patients with COVID-19 [28] through the detection of segmental fibrin thrombi in the glomerular capillary loops related to dysregulation of the coagulation pathway. Other findings supported that thrombotic microangiopathy (TMA) with the formation of thrombi in both pulmonary and kidney blood vessels could induce acute tubular necrosis [29]. In contrast, Mohamed et al. (2020) [30] emphasized that renal microthrombi cannot be included as a remarkable cause of AKI.

Panitchote et al. (2019) [31] postulated that gas exchange impairment and severe hypoxemia can lead to renal medullary hypoxia, and even higher positive end-expiratory pressure (PEEP) in mechanical ventilation may contribute to increased venous pressure and foster kidney congestion. However, this is not in line with our results, as none of the study participants were severely hypoxemic or ventilated.

We excluded patients on nephrotoxic medications from our study, but other authors have referred to AKI as nephrotoxicity related to intravenous iodinated radiocontrast media, or nephrotoxic drugs such as non-steroidal anti-inflammatory drugs (NSAIDs) [32,33].

Other studies have hypothesized rhabdomyolysis, Fanconi syndrome preceding AKI, or imbalanced RAAS activation as a cause of AKI [34,35,36].

While we noticed that AKI was more prevalent among elderly male patients, incidence of AKI was also observed to be more prevalent among patients with hypertension and diabetes mellitus. Patients who developed AKI presented a significantly lower (PO_2_/FiO_2_) and higher SOFA score. These results are consistent with others who have announced that severity of AKI was positively associated with increased age, history of hypertension, diabetes, and higher sequential organ failure assessment (SOFA), which vary in their relationship to AKI risk [31,37].

We found proteinuria (31.3%) and hematuria (19.3%) in COVID-19 patients, and they were more prevalent in the AKI (45.8%, 25%) compared to non-AKI group (25.4%, 16.9%), respectively, which was the same to Naicker et al. (2020) [38], but differed from others [13]. Proteinuria may be linked to direct kidney damage or functional changes caused by fever or infection. However, the coexistence of both proteinuria and hematuria suggests kidney damage rather than functional injury. This is in line with other reports that assumed that proteinuria could be due to a viral-induced cytopathic effect on podocytes and subsequently increased glomerular permeability [39]. This finding has been further supported by Su et al. (2020) [28], who found by transmission electron microscopy that the viral particles in the cytoplasm of the proximal tubular epithelium and podocytes, with podocyte detachment from the glomerular basement membrane and foot process effacement that is associated with proteinuria. Hematuria may occur through direct tubular damage, through intratubular obstruction by blood casts, or microangiopathy [40]. 

Although we did not examine the urine for viral particles, other researchers have reported the presence of SARS-CoV-2 viral nucleic acid particles in the urine samples of COVID-19 patients, suggesting the penetration of viral particles through the glomerular barrier [41]. This further supports potential viral tropism of the kidney, which may be labelled as “COVID-19 nephropathy” [11]. Direct virus invasion of the renal tubular cells and interstitial cells depends upon: (a) ACE2 (Angiotensin converting enzyme-2), expressed in the kidney much more than the lungs, which effectively binds to the S1 domain of the spike protein on SARS-CoV-2. Therefore, ACE2 is considered to be a functional receptor for SARS-CoV-2. (b) TMPRSS2 (Transmembrane Serine Protease 2), which promotes fusion of the viral envelope with cellular membranes with the production of fusion-activated peptides in a step called “priming” [42] (c) CD147, which plays an auxiliary role in SARS-CoV-2 cellular invasion through dysregulated cell cycle and immune-inflammatory responses [43]. Because viral cell passage might include various transmembrane receptors, different receptors may be linked to SARS-CoV-2 kidney disease [44]. SARS-CoV-2 viral variants alpha, beta, gamma, kappa, and delta are spreading with similar levels of infection and this finding has been further supported by Helms et al. (2021) [45], who revealed in a new study that SARS-CoV-2 variants had comparable rates of kidney disease with no statistically significant differences between variants. Furthermore, the patient cohort’s AKI and dialysis rates remained stable throughout the spread of alpha, beta, and gamma variants in the United States [46,47,48,49].

In light of the prevalence of AKI among COVID-19 patients, the factors contributing to its development need to be explored. We used a univariate logistic regression analysis to determine the association between different factors and the development of AKI. Our results revealed that age, hypertension, diabetes mellitus, and a higher SOFA score were identified as independent risk factors for the development of AKI among the studied cases. These findings are in concurrence with previous different reports [3,50,51].

Our study has several limitations: first, the number of patients enrolled in this study was limited, and the duration of observation was not long enough, so there is a possibility of errors. Second, although kidney biopsy can offer better understanding of the histologic pattern of injury and the pathogenic path leading to AKI, we were not able to obtain and present renal pathology data to assess the direct effect of the virus on renal outcomes. Third, we did not analyze the association between kidney impairment and COVID-19 mortality, as there was no available data on the progress of patients after being discharged.

## 5. Conclusions

We thus conclude that AKI may be associated with COVID-19 and has a composite etiology. Several risk factors of AKI were identified. Furthermore, we recommend monitoring the novel biomarkers of kidney injury during hospitalization for COVID-19 patients, which are more reliable in the early prediction of AKI than serum creatinine.

## Figures and Tables

**Figure 1 medicina-58-00443-f001:**
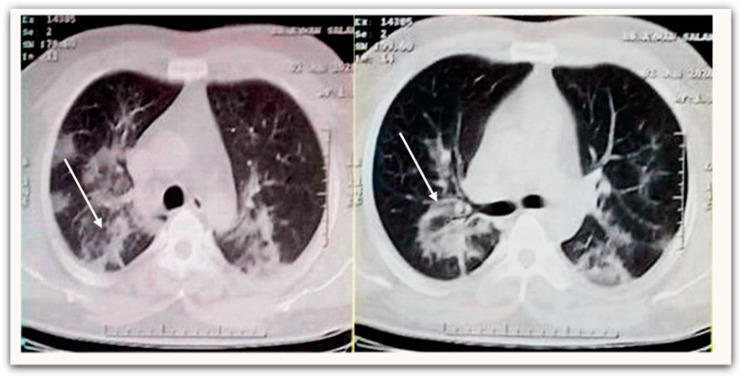
Non-enhanced chest CT scan demonstrates bilateral peripheral ill-defined patchy ground-glass opacities at both lower lung fields (**white arrow**).

**Figure 2 medicina-58-00443-f002:**
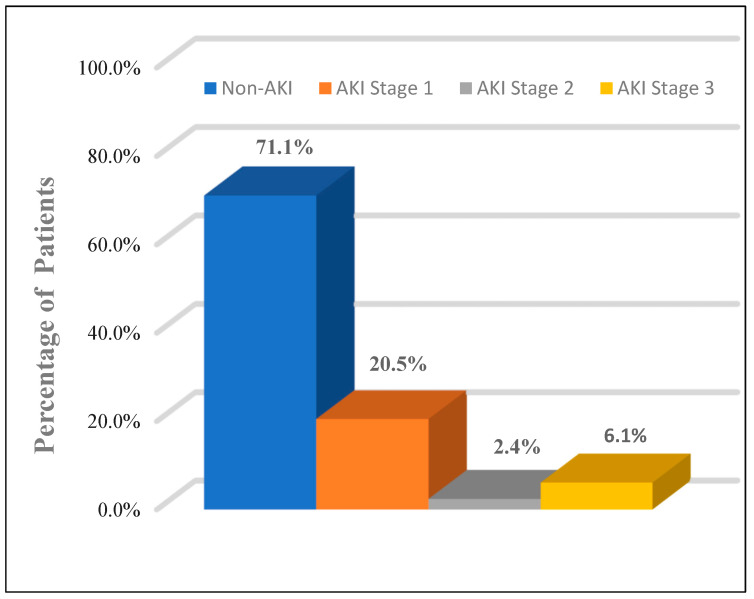
The proportion of patients by stages (1–3) of acute kidney injury (AKI).

**Figure 3 medicina-58-00443-f003:**
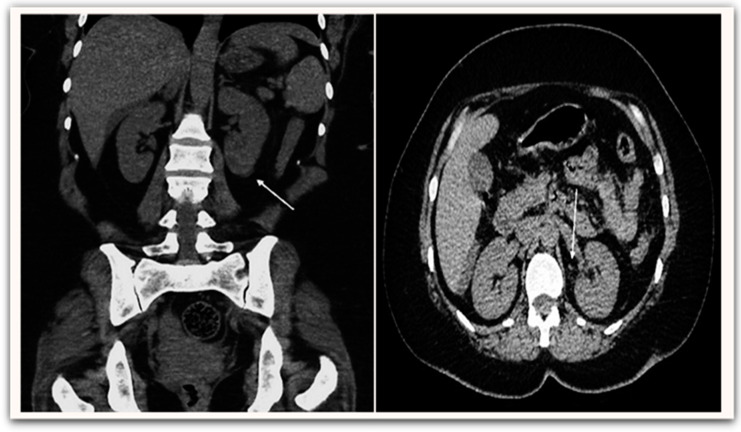
Non-enhanced abdominal CT scan demonstrates perinephric fat stranding (**white arrow**) with irregularity of the outer border of the kidney.

**Figure 4 medicina-58-00443-f004:**
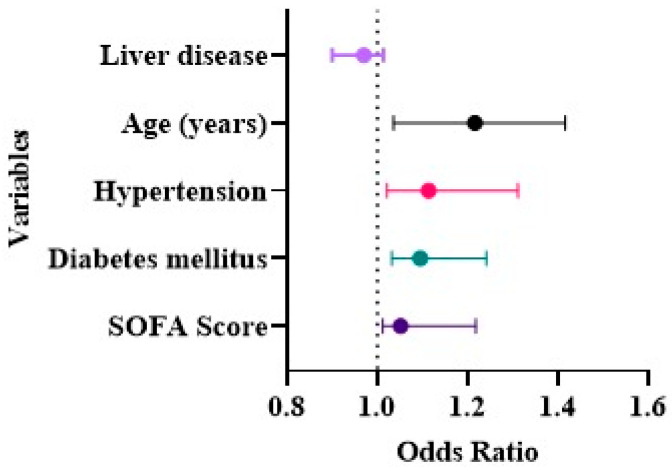
Independent risk factors for (AKI) among COVID-19 patients.

**Table 1 medicina-58-00443-t001:** Staging of acute kidney injury.

Stage	Serum Creatinine	Urine Output
1	1.5–1.9 folds baseline OR ≥0.3 mg/dL increase	<0.5 mL/kg/h for 6–12 h
2	2.0–2.9 folds baseline	<0.5 mL/kg/h for ≥12 h
3	3 folds baseline OR Increase in serum creatinine to ≥4.0 mg/dL OR Initiation of renal replacement therapy	<0.3 mL/kg/h for ≥24 h OR Anuria for ≥12 h

**Table 2 medicina-58-00443-t002:** Sequential Organ Failure Assessment (SOFA) Score.

SOFA Score	0	1	2	3	4
PaO_2_/FiO_2_ (mmHg)	>400	<400	<300	<200 with respiratory support	<100 with respiratory support
Platelet (×10^9^/L)	≥150	100–149	50–99	20–49	<20
Bilirubin (mg/dL)	<1.2	1.2–1.9	2.0–5.9	6.0–11.9	≥12.0
Hypotension	No hypotension	MAP < 70 mmHg	Dopamine ≤ 5 or dobutamine (any dose)	Dopamine > 5, epinephrine ≤ 0.1, or norepinephrine ≤0.1	Dopamine > 15, epinephrine > 0.1, or norepinephrine > 0.1
Glasgow Coma Score	15	13–14	10–12	6–9	<6
Creatinine (mg/dL)	<1.2	1.2–1.9	2.0–3.4	3.5–4.9	≥5.0

**Table 3 medicina-58-00443-t003:** Demographic and clinical characteristics of COVID-19 patients on admission.

Variables	Total Patients	Non-AKI Group	AKI Group	*p*-Value
No. of patients (%)	83 (100.0%)	59 (71.1%)	24 (28.9%)	-
Age (years) (mean ± SD)	52.9 ± 12.4	51.07 ± 12.5	57.71 ± 11.0	-
Sex, *n* (%)	-			
Male	46 (55.4%)	32 (54.2%)	14 (58.3%)	-
Female	37 (44.6%)	27 (45.8%)	10 (41.7%)	-
Comorbid conditions, *n* (%)	-			
Cardiovascular diseases	10 (12.1%)	6 (10.2%)	4 (16.6%)	-
Chronic liver diseases	3 (3.6)	2 (3.4%)	1 (4.1%)	-
Chronic obstructive lung diseases	7 (8.4%)	5 (8.2%)	2 (8.5%)	-
Diabetes mellitus	19 (22.9%)	12 (20.3%)	7 (29.2%)	-
Hypertension	23 (27.7%)	15 (25.4%)	8 (33.3%)	-
Mean arterial pressure (mmHg)	90.1 ± 5.4	90.6 ± 4.2	88.7 ± 7.5	0.222

SD = standard deviation; *n* = number.

**Table 4 medicina-58-00443-t004:** Laboratory indices of COVID-19 patients on admission.

Variables	Total Patients	Non-AKI Group	AKI Group	*p*-Value
Lab Test, Mean (±SD)				
Hemoglobin (g/dL)	12.5 ± 2.1	12.8 ± 2.1	12.1 ± 1.9	0.175
White blood cells (×10^9^/L)	7.8 ± 4.3	7.9 ± 4.2	7.5 ± 4.7	0.635
Lymphocyte (×10^9^/L)	1.2 ± 0.5	1.3 ± 0.6	1 ± 0.3	0.076
Neutrophil (×10^9^/L)	5.9 ± 3.7	5.9 ± 3.6	5.8 ± 3.9	0.908
Platelet (×10^9^/L)	262.4 ± 89.4	283.6 ± 95.8	241.2 ± 83.2	0.216
Albumin (g/dL)	3.8 ± 0.6	3.8 ± 0.6	3.7 ± 0.7	0.359
Total bilirubin (mg/dL)	0.97 ± 0.3	0.9 ± 0.2	1.03 ± 0.3	0.112
Alanine aminotransferase (IU/L)	44.9 ± 22.2	43.6 ± 16.5	48 ± 32.5	0.415
Aspartate aminotransferase (IU/L)	39.5 ± 15.9	38.8 ± 13	41.4 ± 21.5	0.496
Prothrombin time (seconds)	13.1 ± 1.1	13 ± 0.9	13.3 ± 1.4	0.279
Partial thromboplastin time (seconds)	30.5 ± 4.3	30.5 ± 4.5	30.7 ± 3.7	0.811
Ferritin (ng/mL)	582.9 ± 182.3	575.3 ± 169.9	601.1 ± 193.8	0.565
C-reactive protein (mg/L)	8.28 ± 3.13	8.04 ± 3.02	8.87 ± 3.39	0.275
D-dimer (ng/mL)	422.94 ± 75.18	416.53 ± 68.81	438.81 ± 88.54	0.223
Serum Na (mmol/L)	138.6 ± 6.4	139.3 ± 5.9	136.8 ± 6.9	0.107
Serum K (mmol/L)	4.3 ± 0.7	4.23 ± 0.7	4.59 ± 0.7	0.099
Serum Ca (mg/dL)	8.1 ± 0.5	8.12 ± 0.5	8.14 ± 0.5	0.909
Serum Urea (mg/dL)	90.5 ± 50.3	86.5 ± 46.1	100.3 ± 59.4	0.261
Serum Creatinine (mg/dL)	1.7 ± 1.3	1.5 ± 0.8	2.5 ± 1.9	0.007 *
PaO_2_/FiO_2_ (mmHg)	412.02 ± 29.43	416.49 ± 28.46	401.33 ± 29.81	0.029 *
SOFA Score	1.48 ± 0.79	1.33 ± 0.66	1.88 ± 0.95	0.004 *
KDIGO stage				
Stage 1	17 (20.5%)	0 (0)	17 (70.8%)	-
Stage 2	2 (2.4%)	0 (0)	2 (8.3%)	-
Stage 3	5 (6.1%)	0 (0)	5 (20.9%)	-
Proteinuria on hospital admission				
Positive	26 (31.3%)	15 (25.4%)	11 (45.8%)	0.287
Negative	57 (68.7%)	44 (74.6%)	13 (54.2%)	-
Hematuria on hospital admission				
Positive	16 (19.3%)	10 (16.9%)	6 (25%)	0.703
Negative	67 (80.7%)	49 (83.1%)	18 (75%)	-

* Significant difference at *p* ≤ 0.05.

**Table 5 medicina-58-00443-t005:** Risk factors for acute kidney injury using univariate logistic regression analysis.

Variables	Odds Ratio	95% CI	*p*-Value
Age (years)	2.051	1.771–2.373	0.000 *
Cardiovascular diseases	0.993	0.938–1.261	0.294
Chronic liver disease	0.812	0.692–1.494	0.953
Chronic obstructive pulmonary disease	0.741	0.504–1.112	0.151
Diabetes mellitus	1.094	1.032–1.241	0.012 *
Hypertension	1.612	1.424–1.823	0.002 *
SOFA Score	1.051	1.012–1.218	0.023 *

* Significant difference at *p* ≤ 0.05.

## Data Availability

On request, the corresponding author will provide the datasets produced and/or analyzed during the current study.

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
