# Peer review of "Evolving Risk of Acute Kidney Injury in COVID-19 Hospitalized Patients: A Single Center Retrospective Study"

_medicina, 2022, doi:10.3390/medicina58030443_

Round 1

Reviewer 1 Report

Dear authors, I appreciate your manuscript but you have to improve the figure 2 (not clear the color box and relative stage) and to reduce the bibliography that is too long. 

Author Response

  • Comment:

Dear authors, I appreciate your manuscript but you have to improve the figure 2 (not clear the color box and relative stage) and to reduce the bibliography that is too long. 

Answer:

Done. Modified

  • Language Editing

The manuscript was edited by native English speakers.

Reviewer 2 Report

Authors show another important study analyzing risk factors related with AKI and clinical manifestation of AKI in COVID-19 patients. Although seems interesting, the manuscript needs a lot of improvement:

1) in the abstract (line 19) and later I would be careful about statement that AKI is 'rare' in COVID-19 patients, many studies, especially recent metanalysis (Cai X, Wu G, Zhang J, Yang L. Risk Factors for Acute Kidney Injury in Adult Patients With COVID-19: A Systematic Review and Meta-Analysis. Front Med (Lausanne). 2021 Dec 6;8:719472. doi: 10.3389/fmed.2021.719472. eCollection 2021) indicate that AKI is common and serious complication of COVID-19;

2) in the introduction (lines 47-53) please think about removing/modifying whole part about geneaology of COVID-19, it's not related with the manuscript;

3) please think about modyfying sentences in lines 54-59, blood is not the organ; what do you mean by saying that virus 'passes into the blood, gathers in the kidney and causes harm to renal cells'? What kind of tissue damage you're talking about and what kind of cells are damaged? What is the evidence of direct kidney damage? or rather endothelial damage and TMA? please provide more data/references;

4) if your patients had 'laboratory-confirmed COVID-19' (line 74), what kind of symptoms they had? what was their general condition?

5) Was the dose of Vitamin C 1 gram daily? (line 107);

6) What kind of anticoagulation was used when D-dimers were high? (108-109);

7) please use full name of abbreviation used: S.cr in line 115 and later;

8) What does it mean that 'AKI was not diagnosed consistently for the diagnosis of AKI'? (line 120);

9) many disorders in the results section are written in capital letters, please be more specific and explain/give examples what does it mean 'cardiovascular diseases' (line 148) or 'liver disease' (line 149); COPD is not cathegorized as 'lung disease';

10) Do all patients has the same chest CT results? (lines 149-152), image shown in Figure 1 is too general, we don't know which patients data you're using;

11) Similarly, findings on Figure 3 can not be generalized to all patients;

12) Does your study indicate that' increased D-dimers levels in AKI group are related with circulating microthrombi formation'? (lines 234-235), what about thrombi formed in the kidney in situ? what's the evidence for your statement?;

13) please use appropriately the name of the virus: SARS-CoV2 (line 284 and 285);

14) what is an 'association between the different associated factors and AKI'? (line 291);

15) I think that last sentences about relationship between AKI and novel markers (especially lines 307-310) should be removed from conclusions and moved to the discussion, you should analyze that more if you mentioned them and add references.

16) please check the reference style, many cited manuscripts in the Author line have brakes with '...', like missing some Authors or lack of 'et al.'

Author Response

(1.) Comment:

In the abstract (line 19) and later I would be careful about statement that AKI is 'rare' in COVID-19 patients, many studies, especially recent metanalysis (Cai X, Wu G, Zhang J, Yang L. Risk Factors for Acute Kidney Injury in Adult Patients With COVID-19: A Systematic Review and Meta-Analysis. Front Med (Lausanne). 2021 Dec 6;8:719472. doi: 10.3389/fmed.2021.719472. eCollection 2021) indicate that AKI is common and serious complication of COVID-19.

Answer:

There was a debate about the exact relationship between AKI and COVID-19 infection as some studies mentioned that acute kidney injury (AKI) is uncommon in patients with COVID-19 [Wang, L., Li, X., Chen, H., Yan, S., Li, D., Li, Y., & Gong, Z. (2020). Coronavirus disease 19 infection does not result in acute kidney injury: an analysis of 116 hospitalized patients from Wuhan, China. American journal of nephrology, 51(5), 343-348.https://doi.org/10.1159/000507471] and also [Fabrizi, F., Alfieri, C. M., Cerutti, R., Lunghi, G., & Messa, P. (2020). COVID-19 and acute kidney injury: a systematic review and meta-analysis. Pathogens, 9(12), 1052. doi: 10.3390/pathogens9121052]. So due to this conflict, our study aimed to assess the prevalence of AKI among COVID-19 Patients.

(2.) Comment:

In the introduction (lines 47-53) please think about removing/modifying whole part about genealogy of COVID-19, it's not related with the manuscript.

Answer:  Done. Removed

(3.) Comment:

Please think about modifying sentences in lines 54-59, blood is not the organ; what do you mean by saying that virus 'passes into the blood, gathers in the kidney and causes harm to renal cells'? What kind of tissue damage you're talking about and what kind of cells are damaged? What is the evidence of direct kidney damage? Or rather endothelial damage and TMA? Please provide more data/references.

Answer:  See Line 50

Some COVID-19 adverse effects that could contribute to acute renal damage include: increased blood clotting, possible direct infection of the kidney, or damage to kidney cells (or acute tubular necrosis) with septic shock. Recent metanalysis ascertain that the mostly affected renal cells are the Glomerular more than tubular cells and this may explain the associated proteinuria in those patients.

(4.) Comment:

If your patients had 'laboratory-confirmed COVID-19' (line 74), what kind of symptoms they had? What was their general condition?

Answer: See Line 70-75

All studied patients were suspected as COVID-19 patients as they had Fever, Dry Cough, Tiredness, Diarrhea, Loss of Taste and/or Smell. Accordingly, the were isolated in the medical departments. All of them had mild to moderate symptoms of COVID-19 but no evidence of severe pneumonia, such as a SpO2 of less than 90% on room air. So, none of them were admitted to the ICU or underwent mechanical ventilation.

(5.) Comment:

Was the dose of Vitamin C 1 gram daily? (line 107)

Answer: see line 105

Yes, according to Ministry of Health and Population (MOHP), Egypt Management protocol for COVID-19 Patients, version 1.4, 30th May,2020, the recommended daily dose of Vit. C is 1000 mg/day.

(6.) Comment:

What kind of anticoagulation was used when D-dimers were high? (108-109);

Answer: see line 107

Anticoagulation (Rivaroxaban 20 mg daily) if D-dimer > 1000

(7.) Comment:

Please use full name of abbreviation used: S.cr in line 115 and later

Answer: see line 113,114 and Table (1)

Done.

(8.) Comment:

What does it mean that 'AKI was not diagnosed consistently for the diagnosis of AKI'? (line 120)

Answer: see line117-119

Done. Modified

Due to a lack of daily urine measurement in the electronic chart, urine output criteria were not used consistently for the diagnosis of AKI.

(9.) Comment:

Many disorders in the results section are written in capital letters, please be more specific and explain/give examples what does it mean 'cardiovascular diseases' (line 148) or 'liver disease' (line 149); COPD is not categorized as 'lung disease';

Answer: See table (3)

We collectively gathered the comorbidities as general entities such as Cardiovascular diseases (e.g: CAD, Heart failure, cardiomyopathy…etc), liver disease (e.g., liver cirrhosis, viral hepatitis, veno-occlusive disease…etc) and chronic lung diseases (e.g., COPD, Asthma….etc) ,so as to condense the results.

(10.) Comment:

Do all patients have the same chest CT results? (lines 149-152), image shown in Figure 1 is too general, we don't know which patients’ data you're using;

Answer:

Most of the studied patients did not differ much in the chest CT scan, so we took samples of the chest CT images as an example of what we found during the study.

(11.) Comment:

Similarly, findings on Figure 3 cannot be generalized to all patients

Answer:

Most of the AKI Group 24 (28.9%) showed the fat stranding or irregularity of the border of the kidney in different grades. So, we select the most clearly images containing most of details.

(12.) Comment:

Does your study indicate that' increased D-dimers levels in AKI group are related with circulating microthrombi formation'? (lines 234-235), what about thrombi formed in the kidney in situ? what's the evidence for your statement?

Answer: see line 226-227

Done. Modified

(13.) Comment: see line 287-288

Please use appropriately the name of the virus: SARS-CoV2 (line 284 and 285)

Answer: see line 277-278

Done. Modified

(14.) Comment:

what is an 'association between the different associated factors and AKI'? (line 291);

Answer: Done, see line 291-292

 We used a univariate logistic regression analysis to determine the association between different factors and development of AKI.

(15.) Comment:

I think that last sentences about relationship between AKI and novel markers (especially lines 307-310) should be removed from conclusions and moved to the discussion, you should analyze that more if you mentioned them and add references.

Answer: see line 305- 308

(Done. All the paragraph is modified)

We thus conclude that AKI is common in COVID-19 and has a composite etiology. Several risk factors of AKI were identified. Furthermore, we recommend monitoring the novel biomarkers of kidney injury during hospitalization for COVID-19 patients, which are more reliable in early prediction of AKI than serum creatinine

(16.) Comment:

Please check the reference style, many cited manuscripts in the Author line have brakes with '...', like missing some Authors or lack of 'et al.' 

Answer: Done

The bibliography software package EndNote was used to manage references.

Round 2

Reviewer 1 Report

Thanks for your revision

Reviewer 2 Report

Authors answered to all my questions.